# Development in the Mammalian Auditory System Depends on Transcription Factors

**DOI:** 10.3390/ijms22084189

**Published:** 2021-04-18

**Authors:** Karen L. Elliott, Gabriela Pavlínková, Victor V. Chizhikov, Ebenezer N. Yamoah, Bernd Fritzsch

**Affiliations:** 1Department of Biology, University of Iowa, Iowa City, IA 52242, USA; karen-elliott@uiowa.edu; 2Institute of Biotechnology of the Czech Academy of Sciences, 25250 Vestec, Czechia; gabriela.pavlinkova@ibt.cas.cz; 3Department of Anatomy and Neurobiology, The University of Tennessee Health Science Center, Memphis, TN 38163, USA; vchizhik@uthsc.edu; 4Department of Physiology and Cell Biology, School of Medicine, University of Nevada, Reno, NV 89557, USA; enyamoah@gmail.com

**Keywords:** transcription factors, neuronal differentiation, bHLH genes, spiral ganglion neurons, cochlea hair cells, cochlear nuclei

## Abstract

We review the molecular basis of several transcription factors (*Eya1*, *Sox2*), including the three related genes coding basic helix–loop–helix (bHLH; see abbreviations) proteins (*Neurog1*, *Neurod1*, *Atoh1*) during the development of spiral ganglia, cochlear nuclei, and cochlear hair cells. Neuronal development requires *Neurog1,* followed by its downstream target *Neurod1*, to cross-regulate *Atoh1* expression. In contrast, hair cells and cochlear nuclei critically depend on *Atoh1* and require *Neurod1* expression for interactions with *Atoh1*. Upregulation of *Atoh1* following *Neurod1* loss changes some vestibular neurons’ fate into “hair cells”, highlighting the significant interplay between the bHLH genes. Further work showed that replacing *Atoh1* by *Neurog1* rescues some hair cells from complete absence observed in *Atoh1* null mutants, suggesting that bHLH genes can partially replace one another. The inhibition of *Atoh1* by *Neurod1* is essential for proper neuronal cell fate, and in the absence of *Neurod1*, *Atoh1* is upregulated, resulting in the formation of “intraganglionic” HCs. Additional genes, such as *Eya1/Six1*, Sox2, *Pax2, Gata3, Fgfr2b, Foxg1*, and *Lmx1a/b*, play a role in the auditory system. Finally, both *Lmx1a* and *Lmx1b* genes are essential for the cochlear organ of Corti, spiral ganglion neuron, and cochlear nuclei formation. We integrate the mammalian auditory system development to provide comprehensive insights beyond the limited perception driven by singular investigations of cochlear neurons, cochlear hair cells, and cochlear nuclei. A detailed analysis of gene expression is needed to understand better how upstream regulators facilitate gene interactions and mammalian auditory system development.

## 1. Introduction

The mammalian auditory system evolved in tetrapods out of the vestibular system [1,2] that diversifies into distinct hair cells with a unique innervation of spiral ganglion neurons (SGNs) which reach the cochlear nuclei for information processing [3,4]. The development of the auditory system depends on sequential activation of various genes. Once early development is completed, age-dependent loss progresses to reduce auditory hair cells that cannot be replaced [5,6]. Loss of hair cells may result in a delayed loss of SGNs with age [7,8], which will affect the cochlear nuclei [9]. We aim to provide an overview of transcription factors to allow a framework to develop the auditory system’s earliest steps, from neurons to hair cells and cochlear nuclei.

The mammalian auditory system consists of SGNs, two types of cochlear hair cells (inner and outer hair cells; IHC, OHC), and the cochlear nuclei subdivided into three nuclei: the anteroventral, posteroventral, and dorsal cochlear nucleus. Insights into these interlinked neurosensory components will be detailed beyond the initial morphogenetic early steps [9,10]. The proliferation pattern within each auditory system component affects the temporal and spatial differentiation and interconnects with various neurosensory cells. SGNs are the first cells of the auditory system to exit the cell cycle, progressing from base-to-apex from ~E10.5–12.5, followed by cochlear HCs, which exit the cell cycle opposite direction, from apex-to-base between ~E12.5–14.5 [11,12]. The temporal pattern of proliferation progression in the cochlea raises questions: why do developing neurons proliferate from the apex-to-base, and the cochlear HCs proliferate from base-to-apex (Figure 1)? The cochlear nuclei exit the cell cycle between E10.5–14.5 [13], partially overlapping with spiral ganglion neurons. Granular cells exit the cell cycle between E12-18, which is delayed relative to spiral ganglion proliferation progression [14]. We want to follow the formation of SGNs, hair cells, and cochlear nuclei and relate them to the three interlinked connections.

Spiral ganglion neurons: SGNs depend upon *Eya1* and *Sox2,* followed by *Neurog1* [15] and *Neurod1* [16]. *Neurog1* was shown to be regulated by *Neurod1* [17]. In contrast to the complete loss of neurons in *Neurog1* null mice [15], a few SGNs remain following the loss of *Neurod1* [16,18]. A degree of interaction and crosstalk between *Neurog1, Neurod1*, and *Atoh1* and various additional transcription factors remain incompletely understood and will be discussed in this review.

**Figure 1 ijms-22-04189-f001:**
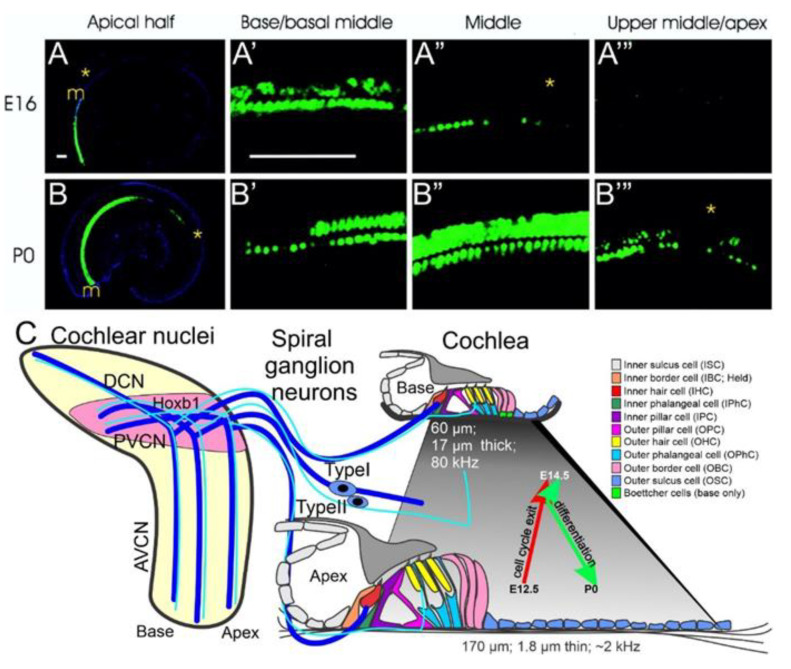
A composite set of development of the auditory system is revealed—Base-to-apex progression of α9-AChr expression in differentiating hair cells along the cochlea (**A**,**B**). The two different types of spiral ganglion neurons connect the cochlear hair cells with the cochlear nuclei (anteroventral (AVCN), posteroventral (PVCN), dorsal (DCN)), maintaining basal to apical tonotopic representation (**C**). The PVCN, located in rhombomere 4, expresses Hoxb1. An apex-to-base proliferation (E12.5–14.5) for cochlear hair cells is followed by a base-to-apex differentiation initiated by Atoh1 (E13.5–P0). The organ of Corti contains two different types of hair cells, inner and outer hair cells (IHC, OHC) arranged from high-frequency detection at the base (thick basilar membrane) to low-frequency detection at the apex (thinner basilar membrane). Approximately 15 type I fibers converge on a single IHC, whereas a single type II fiber expands along several OHCs. Central projections of type I and type II fibers are mostly parallel in the AVCN, maintaining a topographic map, though a slightly different pattern of organization exists in the PVCN. Modified after [9,19,20,21].

Cochlear hair cells: The unusual progression of cell cycle exit from apex to base and subsequent differentiation from base to apex in cochlear HCs has been studied by gene expressions, including *Eya1, Sox2,* and *Atoh1*. The gene, *Cdkn1b* (*p27^(kip)^*), showed a distinct progression from the apex to the base [22,23] and is a marker of cell cycle exit followed by differentiation (Figure 1). About the same time, it was shown that cochlear HCs depend on the bHLH gene *Atoh1* (*Math1*), as *Atoh1* null mice lacked all HC differentiation [24]. *Atoh1-eGFP* mice showed a clear base to apex progression of *Atoh1* expression [25], showing a different progression compared to cell cycle exit. Another bHLH null mutant, *Neurog1*, showed an altered pattern of cochlear HCs: the presence of up to 2 rows of IHCs and 4 to 5 rows of OHCs near the apex, a complete absence of neurons, and a reduced cochlear length of ~50% [15] suggesting that *Neurog1* is upstream of *Atoh1* expression [26,27]. A similar effect of cochlear length reduction was observed following the loss of another bHLH gene, *Neurod1*. In these mice, the length of the cochlea was reduced by about 1/3 and showed alterations of HCs: some OHCs showed an “IHC”-like phenotype [18,28]. *Neurod1*, like *Atoh1*, is also downstream of *Neurog1* [17], suggesting an interaction of these bHLH genes with each other [27,29]. Finally, HC differentiation from the base to the apex was first described using *Chrna9* (*α9-acetylcholine)-eGFP* expression [19]. It showed that HCs began differentiating first near the base around E16 and progressed over time toward the apex (Figure 1). Furthermore, expression begins in IHC before its expression in OHC, consistent with *Atoh1* expression that contrasts with the proliferation of cell cycle exit from apex to base [30,31], including *Nmyc2* [32].

Cochlear nuclei: *Eya1* and *Sox2* are expressed in the cochlear nuclei that requires the upregulation of *Atoh1* for cochlear nuclei development [33]. Using *Gata3* LacZ expression in delaminating and differentiating SGNs, it was established that these neurons reach the cochlear nuclei starting at E12.5 [34,35]. No cochlear projections ever form in *Neurog1* null mice, given the loss of all SGNs [15]. The claim of central projection loss in *Neurod1* null mice [18] was overstated as it was subsequently shown that the central projections of SGNs were formed and reached the cochlear nuclei [16,36]. In contrast to the possible transient expression of *Neurog1* in cochlear nucleus neurons, there was a massive expression of *Neurod1* [37], overlapping with *Atoh1* expression [33] in the cochlear nuclei. In the cerebellum and the cochlear nucleus, *Neurod1* negatively regulates *Atoh1* expression [38]. Later work showed discrete expression of *Atoh1* and *Ptf1a* in cochlear nuclei progenitors: *Atoh1* is expressed in the progenitors of the excitatory cochlear nuclei neurons *Ptf1a* in the progenitors of the inhibitory cochlear nuclei neurons [39]. Simultaneously, it has been shown that expression of *Atoh1* in progenitors that predominantly populate dorsal cochlear nuclei depends on secreted signals regulated by *Lmx1a* and *Lmx1b* in the hindbrain roof plate [40].

This review seeks to compare various aspects of the three different neurosensory components: the spiral ganglion neurons (SGN), the cochlear hair cells (HCs), and connect them to cochlear nuclei. We provide a comprehensive overview to interlink the spiral ganglia, cochlear HCs, and cochlear nuclei in the context of regulation by bHLH (*Neurog1*, *Atoh1*, and *Neurod1*) and other transcription factors (*Eya1/Six1*, *Sox2*, *Pax2*, *Gata3*, *Lmx1a/b*) to understand the development and aging processes of related neurosensory cells and their central projections to HCs and cochlear nuclei. Gene regulation in the SGNs, HCs, and cochlear nuclei is critical for long-term viability and potential regeneration of lost neurosensory elements [5,6,41]. What remains to be demonstrated is how the different genes interact within the mammalian auditory system’s various components.

## 2. Neurog1 Regulates Neurod1 and Atoh1 Expression and Is Essential for Spiral Ganglion Neurons

The absence of *Neurog1* resulted in a reduced length of the organ of Corti and prevented the development of all sensory neurons (Figure 2A–C), as they never initiated neuronal differentiation [12,17]. In contrast, *Neurod1* null mice did not eliminate all sensory neurons [16]. Loss of *Neurod1* resulted in a prolonged *Atoh1* expression in sensory neurons and differentiation of some of these *Atoh1*-positive cells into ectopic “intraganglionic” HCs [28]. Furthermore, *Neurod1* elimination in differentiating SGNs resulted in the formation of an abnormal spiral-vestibular ganglion [42,43], including the formation of vesicles containing several *Atoh1*-positive “HCs” [28,44]. These neurons overlap with the transient *Atoh1*-positive cells, suggesting *Atoh1* is typically suppressed by *Neurod1*, consistent with the regulation of *Atoh1* in the cerebellum [38] and the intestine [45]. Mice null for both *Atoh1* and *Neurod1* do not form “intraganglionic” HCs as was observed in mice lacking only *Neurod1* [43].

SGNs are dependent on neurotrophins (Figure 2D–F). Loss of *Ntf3* results in a complete loss of neurons innervating the cochlea base, whereas loss of *Bdnf* results in fewer neurons innervating the apex [48,49]. The loss of both copies of *Ntf3* and one copy of *Bdnf* further reduces innervation to the base (Figure 2G). However, misexpression of *Bdnf* from the *Ntf3* promoter causes an exuberant projection of neurons to the basal turn (Figure 2H), resulting in redirecting vestibular fibers to innervate the cochlea broadly [47]. Loss of both neurotrophin receptors (*Ntrk2* and *Ntrk3*; [50]) or both neurotrophins (*Bdnf/Ntf3*) results in the complete loss of all neurons that leads to a subsequent loss of HCs [8].

SGNs can be classified as type I neurons, which innervate the IHCs, and type II neurons that innervate OHCs (Figure 1 and Figure 3). The latter is a much smaller proportion, constituting only 5-7% of the total spiral ganglion neurons. Currently, we do not fully understand the molecular mechanisms regulating the development, segregation, and neuronal projections of SGNs, although several genes involved in these processes are known [51,52,53,54,55]. For example, in *Prox1* mutants, type II fibers turn at random and merge into a single bundle compared to the three parallel fibers that turn toward the base in controls (Figure 3A,B; [30,56]). Furthermore, the loss of Schwann cells following *Sox10* deletion results in fibers projecting beyond the target HCs (Figure 3C; [54,57,58]). Additional genes, such as the Wnt/planar cell polarity (PCP) genes, *Vangl2*, *Fzd3,* and *Fzd6*, have been shown to affect type II fiber guidance to OHCs [58].

Mice in which *Atoh1* was replaced by *Neurog1* (*Atoh1^kiNeurog1^*) showed a different inner ear innervation pattern. In this model, there is an increased density of innervation to undifferentiated HCs [59,60]. Conditional deletion of *Neurod1* results in incomplete neuronal loss, and the remaining fibers formed aberrant central and peripheral projections [36,42]. Dense neuronal projections to the flat epithelia were found in *Atoh1/Neurod1* null mice in the absence of HCs [43], suggesting an uncoupling of innervation and HC differentiation.

In summary, the interrupted development of HCs in double *Atoh1/Neurod1* null mutants uncouples HC development and projections growth. Expanded innervation of the remaining neurons in mouse mutants [43] could help hearing loss restoration [61], as well as the formation of neurons upstream of the proliferation of otic progenitors [62]. Regeneration of SGNs is also an essential task for future studies [62].

### 2.1. Is Atoh1 Playing a Role in Spiral Ganglion Neurons

Analysis of *Atoh1-eGFP* transgenic mice or *Atoh1-cre* allele showed transient expression of *Atoh1* in some SGNs [12]; some of these neurons retained long-term expression *Atoh1* [12]. These data suggest that some precursors may give rise to both HCs and sensory neurons. More substantial support of this hypothesis comes from recent data using multicolor labeling, which indicates a clonal relationship between several different inner ear cell types, including some sensory neurons [63]. It was later shown that the ordinarily transient expression of *Atoh1* in inner ear sensory neurons could be sustained following the loss of *Neurod1* [28], suggesting that *Neurod1* inhibits *Atoh1* expression, consistent with crosstalk of these two bHLH genes. In *Neurod1* null mice, *Atoh1* becomes upregulated in a different pattern (apex to base), also allowing “neurons” in the vestibular and cochlear ganglions to differentiate as ectopic “HCs” that were often grouped around intraganglionic vesicles [28,42,43]. The cochlear-derived ectopic HCs follow transient expression of *Atoh1* that interacts with *ErbB2,* and *Sox10* derived Schwann cells [57,64]. Without neuron formation, we see a delay of *Atoh1* expression (Figure 4A,B), including different disorganized central projections in near-complete absence of IHCs, using exuberant type II projections [44,65].

Hair cells depend on neurons for their long-term maintenance [8], and conversely, neurons depend upon HCs and supporting cells [9,47]. Logically, one would assume that the absence of HCs will cause degeneration of eventually all neurons due to missing neurotrophic support. While the initial targeted growth of afferents and efferents to the undifferentiated “HCs” was not affected in *Atoh1*-null mice, there was indeed a later loss of innervation to certain inner ear areas [68]. Interestingly, the areas of sustained innervation by neurotrophins correlated with areas that still expressed *Bdnf* and *Ntf3* in the absence of *Atoh1* protein, suggesting that fiber retention is possible in the absence of *Atoh1* differentiated HCs. Only immature HCs are formed in *Pou4f3* null mice, and gradually HCs disappear in these null mice with time, and yet, some of the afferents are retained for a long time in these mice with limited expression of *Bdnf* and *Ntf3* [70,71]. Furthermore, various *Atoh1* conditional deletion mutations show a residual partial formation of HCs and proportional loss of neurons [43,72,73]. Finally, conditional deletion of *Neurod1* results in a loss of many SGNs, yet near-normal cochlear HCs developed [43,68], suggesting that HCs’ survival depends on SGNs that require neuronal connections for their neurons.

In summary, neurons depend on HCs and vice versa. Understanding how the expansion of neuronal projections happens in the absence of HCs could restore lost innervation [61] and allow the flat epithelial to go beyond the expansion in the absent HCs [43,74]. Transforming SGNs to develop as HCs would provide a new population of HCs that adds to a possible source of HCs [5,28].

### 2.2. Spiral Ganglion Neurons Depend Upon Eya1 and Sox2 and Other Genes

*Eya1* is expressed before *Sox2,* and both play a critical role in the proliferation and neurogenesis of neurosensory cells, and their loss will eliminate all sensory neurons [75,76,77]. Conditional deletion of *Sox2* by *Foxg1-cre* likewise showed the absence of spiral ganglia but permitted a transient development of vestibular neurons, including developing peripheral and central processes [78]. A similar lack of SGNs is demonstrated in deleting *Eya1/Six1* following the loss of *Brg1* expression, indicating a major role for SWI/SNF chromatin remodeling [77]. Furthermore, a delayed deletion of *Sox2* using *Isl1-cre* resulted in a complete loss of HCs and sensory neurons in the apex but had a transient expansion of disorganized HCs, with an unusual base innervation [79]. The similarities and differences between the different *Sox2* deletions (*Lcc, Ysb, Isl1-cre; Foxg1-cre*) remain incompletely analyzed [78,80].

*Eya1* elimination results in all neurons’ obliteration in the inner ear [63,77,81]. Also, *Pax2*-null mice resulted in a near absence of SGNs except for a small network of neurons innervating the cochlear duct [66], comparable to the partial innervation *Isl1-cre; Sox2^f/f^* mice [79]. Deleting *Pax2* also results in the cochlear sack’s prolapse into the braincase beneath the brainstem (Figure 4H; [66]). Various other genes either entirely or partially derail the process of innervation [30,51,58,82]. For example, deleting even a single neurotrophin (*Bdnf* or *Ntf3*) results in some neuronal loss and aberrant innervation of the remaining neurons’ sensory epithelia [46,47]. Other genes, such as *Gata3* and *Dicer*, affect inner ear neurosensory development [83,84]. Conditional deletion of either *Gata3* or *Dicer* by *Atoh1-cre* results in a partial loss of neurons and HCs [83,84]; however, conditional deletion of *Gata3* or *Dicer* by *Foxg1-cre* results in the complete or near-complete absence of neurons and HCs in a cochlear duct [83,85].

Similarly, in *Lmx1a/b* double knockout (DKO) mice, the cochlea exists as a simple sac devoid of HCs and SGNs, whereas the vestibular neurons and vestibular HCs do form [67]. Finally, even if there is no loss of neurons, Schwann cells’ absence significantly affects neuronal guidance. *Wnt1-cre; Sox10^f/f^* mice lack Schwann cells and had unusual SGN migration and projections to the cochlea’s lateral wall, bypassing the organ of Corti (Figure 3C; [57]), suggesting that Schwann cells provide a stop signal for SGNs.

In summary, the deletion of specific genes results in either complete neurosensory loss or some residual and transient sensory neuron formation. At least two and possibly five bHLH genes [29] interact to facilitate regular *Atoh1* expression (Figure 2, Figure 3 and Figure 4). Among these genes are *Neurog1* and *Neurod1,* which interact before *Atoh1* upregulation occurs [27].

## 3. Cochlear Hair Cells Require Atoh1

The original study of *Atoh1* in the inner ear showed that all HCs critically depend upon *Atoh1* expression: without *Atoh1* expression, all HCs are absent [24]. A follow-up study showed that the specific role of *Atoh1* shows undifferentiated *Atoh1-LacZ*-positive HCs [25]. *Atoh1* expression progresses around E13.5 from the upper-middle turn bilaterally toward the base and apex and finishes in the apex between E17–E18 [25,68]. Labeling with *Atoh1*-LacZ shows the latest forming “undifferentiated” HCs in the apex (Figure 4B) [24,68]. Expression of ”undifferentiated” cochlear HCs was near the cochlear IHCs positive cells expressed *Atoh1-LacZ* and demonstrated the loss of HCs at E13.5–E18.5 [25,68]. Furthermore, in situ hybridization showed expression of *Atoh1* in the middle turn at E13.5 [12,28].

Using a short-term BrdU exposure, HCs were shown to exit the cell cycle at E12.5 in the apex and progress toward the base (Figure 1 and Figure 2) [12,86]. In addition, cell cycle exit progresses radially from IHCs to OHCs [28,31,44,87,88]. Questions remain regarding HC maturation progression from base to apex [19]. How the coordination of different cell cycle exits and cell fate determination occurs remains unclear, particularly in the *Atoh1* expressing non-HCs, such as inner pillar cells [12,89]. Specifically, the cellular processes driving remodeling of the prosensory domain during cochlear development suggest combinations of cellular growth and cochlear extension from base-to-apex, allowing a different OHC progression for interpretation [31,44,87,90].

In parallel, various ways of analyzing the cellular progression focusing on *Atoh1* deletions showed an increase in late forming apical HCs. They demonstrated loss of *Atoh1* throughout the organ of Corti [68] correlated with cell apoptosis in identifiable, undifferentiated IHCs and OHCs. Likewise, conditional deletion of *Atoh1* using *Pax2-cre* showed that most HCs were lost during late embryonic development; however, some undifferentiated cells expressed *Myo7a*, a marker for differentiated hair cells postnatal stages, and were innervated by neurons [72]. A ‘self-terminating system (*Atoh1-cre; Atoh1^f/f^*), in which there is a transient expression of *Atoh1* resulting in some initial HC development, demonstrated progressive loss of IHC and most OHCs shortly after birth [73]. These data suggest that a brief upregulation of *Atoh1* suffices to induce differentiation of IHCs (Figure 5C,F) and mostly the first row of OHCs. *Atoh1* deletion at different HC differentiation stages showed different effects on HC survival and stereocilia development depending on the deletion time [91,92], suggesting that *Atoh1* is necessary throughout HC differentiation. Using an ingenious expression system, in which induction of *Atoh1* is under the control of a tetracycline-response element, generated viable ectopic “HCs” adjacent to the organ of Corti in early postnatal mice, with characteristics consistent with endogenous HCs [93], and in line with an upper limit of the proliferation of later HCs [94], consistent with a delay formation of HCs that remain *Myo7a*-positive in undifferentiated HCs.

Finally, loss of IHCs in *Srrm4* mutants (a serine/arginine repetitive matrix 4 genes referred to as *bv/bv* mutations) shows that most type I afferent fibers reroute to OHCs (Figure 5B,G; [44,65]). Notably, the expression of dominant-negative *REST* in double *Srrm3/4* mutants ablates all IHCs [95]. Previous work showed that delayed HCs loss, such as in *Bdnf* and *Ntf3* double CKO mutants, results in delayed innervation [8]. The latter findings are consistent with reducing neurotrophins in *Pou4f3* mutants, resulting in delayed neurosensory loss [70,71].

In mammals, HCs do not regenerate as they do in other vertebrates; however, supporting cells can be induced to transdifferentiate into HCs [5,94,96,97,98,99,100]. Despite this tremendous initial success of HC induction in vitro [41,101,102], reliable generation of new cochlear HCs has not yet been achieved to replace lost HCs, in particular, in older adults [6,103]. Attempts are underway to induce new hair cells by combinations of several transcription factors [104]. Given that *Eya1* regulates *Sox2* that depends on *Atoh1*, no attempt has been made to combine these three transcription factors [77,78].

### 3.1. Neurog1 and Neurod1 Regulate Atoh1 in Cochlear Hair Cells

Among the several bHLH genes known to participate in neurosensory development (*Neurog1, Neurod1, Nscl1, Nscl2, bHLHb5*), we selected a set of genes that have an apparent effect on the cochlea when deleted [15,28,105]. Loss of *Neurog1* in mice results in a premature upregulation of *Atoh1* in an apex-to-base progression and early differentiation of HCs [12,15]. In these mice, HCs exit the cell cycle in an apex to base progression about two days earlier than controls [12], reducing the cochlear length (Figure 4C). Deletion of *Neurod1* also reduces cochlear length (Figure 4A,B) but is less reduced compared to *Neurog1* null mice [18,36,42]. In contrast, *Atoh1* deletion has minimal effect on cochlear length, either by itself or when deleted together with *Neurog1* (Table 1). However, *Neurod1/Atoh1* double null showed an additional reduction relative to that observed in *Neurod1* null mice [43], suggesting a unique interaction of *Atoh1* and *Neurod1*. Similar to *Neurog1* null mice, premature differentiation HCs were found in *Neurod1* mutants (Figure 4C). In these mice, there was an early upregulation of several genes, such as *Atoh1* and *Fgf8*, in the apex where ectopic IHC-like cells in the region of OHCs were found, suggesting these gene proteins interact with each other to specify HC subtype as well [18,28,42,73]. Consistent with *Neurod1* and *Insm1* in the pancreas [106], we suggest the same interactions with OHC to transform into IHCs [28,42,107,108].

Replacing *Atoh1* with *Neurog1* (*Atoh1^kiNeurog1/kiNeurog1^*; Figure 5D,H) resulted in a few patches of undifferentiated “HCs” beyond the flat epithelia of *Atoh1* null mice [59], suggesting that *Neurog1* can partially rescue the *Atoh1* phenotype. The co-expression of both *Neurog1* and *Atoh1* in *Atoh1^+/kiNeurog1^* heterozygotes resulted in disorganization of HC polarity and stereocilia. It was not observed in *Atoh1* heterozygote mice, suggesting that *Neurog1* expression, and not *Atoh1* haploinsufficiency, disrupts HC organization [59]. Combining the novel *Atoh1* self-terminating mouse with the *Atoh1^kiNeruog1^* mouse (*Atoh1-cre*; *Atoh1^kiNeurog1^/Atoh1^f/f^*) showed significantly more differentiated HCs and a more prolonged rescue than in the *Atoh1* “self-terminating” littermates [60]. It is essential instead of transiently rescuing a few apical HCs as in the *Atoh1* “self-terminating” mutant (Figure 5C,F; [73]), significantly more HCs differentiated and persisted for up to nine months in the *Atoh1-cre*; *Atoh1^kiNeurog1^/Atoh1^f/f^* mutant mouse model [60]. Together, these results suggest that, while *Neurog1* can partially replace *Atoh1*, it cannot fully compensate for it. Thus, although replacing *Atoh1* with fly ortholog *atonal* rescued the HC differentiation [109], the subsequent duplication and diversification of *atonal* family bHLH genes [110] no longer allow for the substitution of one for another to ultimately rescue hair cell loss.

These data suggest that the crosstalk of Neurog1, Neurod1, and Atoh1 affects cochlear extension and HC morphology and patterning (Figure 5I; [44]). Fully understanding the various mutations and putting them into the context of different cell fates requires additional work [111,112,113].

### 3.2. Eya1 and Sox2 and Other Transcription Factors Are Essential for HC Development

*Eya1* is regulated by *Sox2*, which governs the *Atoh1* gene [80,114], vital for HC formation [80]. Interestingly, two independent approaches used a delayed deletion of *Sox2* [79,114] but showed different results. In one study, a delayed loss of *Sox2* demonstrated effects in the apex only [114]. In the other study, conditional deletion of *Sox2* resulted in the loss of HCs in the apex and a delayed loss in the base, showing unusual basal turn HCs/supporting cells and inner pillar cells [79]. Timing of *Sox2* expression was demonstrated to be important for sensory development [76,115]. Furthermore, a complete deletion of *Sox2* using *Foxg1-cre* demonstrated the overall cochlear reduction while showing no HC development [78]. These works provide the essential roles of *Sox2*, though the interaction of *Sox2* with *Atoh1* is not fully understood [87,90,99,100].

Other genes are also essential for cochlear development. For example, *Eya1/Six1* are critical for early ear development and are needed to form the cochlea [77,81,116,117]. Another gene, *Pax2*, is necessary for the organ of Corti formation [66] and cooperates with *Sox2* to activate *Atoh1* expression [118]. Conditional deletion of *Gata3* using *Pax2-cre* showed an incomplete loss of HCs compared to the complete absence of HCs using *Foxg1-cre* [34,83]. In *Foxg1-cre; Gata3^f/f^* mice, levels of *Atoh1* expression were significantly reduced, and genes downstream of *Atoh1* were not detected. Mice mutant for *Lmx1a* shows a delayed *Atoh1* expression (Figure 4E–G) followed by transforming some organ of Corti HCs into differentiated vestibular hair cells [69,119,120,121]. In addition, *Lmx1a* deletion results in a shortened cochlea (Figure 4). Similarly, *Foxg1* null mice also show a reduced cochlear length and a disorganized apex containing multiple rows of HCs with disoriented polarities [122]. Furthermore, HC survival is shortened in *Foxg1* null mice [123]. A somewhat similar phenotype is reported for *Nmyc2* null mutants, accompanied by apical cell fate changes [1,32].

The partial deletion of some HCs, but not others, following various gene mutations is an exciting perspective that needs to be explored. For example, the elimination of *Fgfr1* by *Foxg1-cre* in the inner ear epithelium leads to a drastic reduction in the number of auditory HCs found in sensory patches [124]. Likewise, *Sox2* deletion shows a similar partial loss of HCs in the *Yellow submarine* (*Ysb*) mutant [80]. Using *Pax2-cre* to delete *Dicer* [84] conditionally or *Gata3* [83] resulted in incomplete HC loss compared to *Foxg1-cre* conditional deletion of these genes [83,85]. Like the latter conditional mutants, *Lmx1a/b* null mice show a complete loss of all cochlear HC development [67]. These data indicate that cochlear HCs are affected by single deletions and complex interactions of several genes. To date, the effects of compound deletions remain mostly unexplored [101,125]. *Atoh1* may be the dominant gene in HC development [24], but interactions with other genes need to be better understood [99,117].

In summary, various genes upstream of *Atoh1* are essential for HC development, including *Eya1, Sox2, Pax2, Foxg1, Gata3, Lmx1a,* and *Fgfr1,* as their deletion results in aberrant HC formation. Focusing on the effects of early gene deletions with massive HC loss or complete depletion of *Atoh1* will help understand normal HC development [5], possibly using combinations of *Eya1*, *Sox2,* and *Atoh1* to induce new hair cells [77,78].

## 4. Cochlear Nuclei Depend on Atoh1

*Atoh1* deletion disrupted the spinal cord, brainstem, and cerebellum development [126,127] and resulted in loss of all cochlear HCs [33,37,128,129]. Rhombomere-specific deletion of *Atoh1* demonstrated that the cochlear nucleus is formed from cells in rhombomeres 3–5 [129,130]. Furthermore, loss of *Atoh1* or another bHLH gene, *Ptf1a*, resulted in a loss of excitatory or inhibitory cochlear nucleus neurons, respectively, signifying their importance for regulating cell fate determination [39,131]. Moreover, *Lmx1a and Lmx1b* LIM-homeodomain transcription factors expressed in the hindbrain roof plate [40] together regulate *Atoh1* expression [67]. *Lmx1a/b* double null mice lack cochlear development and excitatory neurons in the auditory nuclei, most likely due to the lack of *Atoh1* in these mice [67]. In *Lmx1a/b* double mutant mice, roof plate does not differentiate into the choroid plexus, which is associated with lack of expression of roof plate/choroid plexus-derived secreted molecules, such as *Gdf7* and *Bmps*, known to induce *Atoh1* expression in adjacent neuroepithelium [132,133,134].

Central projections of inner ear afferents of *Atoh1* null mice show near-normal projections, despite the absence of differentiated HCs and cochlear nucleus neurons [135]. Moreover, conditional deletion of *Atoh1* in the ear, but retaining *Atoh1* expression in cochlear nuclei, show near-normal segregation of central projections [43], expanding the critical independence of *Atoh1* in neuronal pathfinding (Figure 6). In contrast, central projections are highly disorganized in *Neurod1* mutants [36,42]. These data suggest that inner ear afferents rely more on molecular cues in the region of the targets than on direct interactions with target cells.

In summary, *Atoh1* regulates cochlear nuclei formation [9,33,37], but has no apparent role in central projections’ guidance [43,135]. More work is needed to understand the role of *Atoh1* to regulate central projections fully.

### 4.1. Neurod1 Interacts with Atoh1 in the Cochlea

Beyond a transient and limited expression of *Neurog1* in vestibular nuclei [37,136], other bHLH genes are expressed in cochlear nuclei (Figure 7 [110,137]). In addition to *Atoh1*, *Neurod1* and *Neurod4* are also expressed in cochlear nuclei [37,136]. *Neurod1* and *Atoh1* negatively interact in the cerebellum [38], in the cochlear HCs and neurons [28], and the intestine [45]. Also, differential levels and expression patterns of *Atoh1* and *Neurod1* are found in the cochlear nucleus [38,138]. For example, *Atoh1* is highly expressed in the ventral cochlear neurons, whereas *Neurod1* is expressed more prominent in the dorsal cochlear nucleus (Figure 8; [18]). An additional bHLH transcription factor, *bHLHb5* [130], is also required to form the dorsal cochlear nucleus properly. Its expression overlaps with yet another bHLH gene, *Ptf1a,* in the dorsal cochlear nucleus [131]. These data implicate the interaction of different bHLH genes in forming the cochlear nuclei (*Atoh1, Neurod1, Ptf1a, bHLHb5*) to crosstalk with each other in the spinal cord [139].

Loss of *Neurod1*, and likely *Neurod4* [136], negatively affects the central targeting and branch pattern of fibers, leading to highly aberrant central pathfinding of both auditory and vestibular afferents [36,42]. Likewise, the inferior colliculi projections are disorganized [42], expanding previous work with *Hoxb2* mutants [140]. Not surprisingly, *Atoh1/Neurod1* double null mice had more severely disorganized projection of cochlear afferents [43] beyond that observed in *Neurod1* single null mice (Figure 6) and *Atoh1* null mice that lack a similar phenotype [135].

In summary, intricate interactions of different bHLH genes (Figure 8) to each other and with *Atoh1* drive the formation of central projections [127,128,134] and, thus, require additional detailed analysis.

### 4.2. Sox2 Function Is Unclear, but Lmx1a/b Double Null Mice Eliminate All Cochlear Nuclei

*Sox2* is essential for proneuronal regulation in the entire brain [77,141,142,143] and is broadly expressed in cochlear nuclei; however, the effects of selective *Sox2* deletion in cochlear nuclei have not yet been analyzed. *Eya1* is expressed in the rhombomere [144] but has not been analyzed selectively; targeted deletion of *Eya1* could clarify its role for the cochlear nuclei. Using *Foxg1-cre* to eliminate *Sox2* in the inner ear results in a complete loss of spiral ganglion formation but does not initially affect vestibular ganglion development [78].

As mentioned above, *Lmx1a/b* double null mutants lack cochlear nuclei [67]. While *Lmx1a* mutants show near-normal central projections [67], spiral ganglion projections of *Lmx1a/b* double null mutants are lost, and vestibular fibers project bilaterally to the dorsal hindbrain and interdigitate with contralateral vestibular fibers (Figure 7 and Figure 8) [67]. The presence of bilateral vestibular projections correlated with other genes’ expression, such as *Wnt3a*, which in *Lmx1a/b* double mutant becomes ectopically expressed at the dorsal midline than in the rhombic lip [67]. The suggested *Wnt3a* attraction expands on previous data showing that loss of the Wnt receptor, *Fzd3* [145], or downstream Wnt signaling component, *Prickle1* [82], affects central projections. Recent work suggests that another gene, *Npr2*, affects central projections, showing the gain and loss of afferents to different cochlear nuclei [146]. Central projections in mutants for other genes (*Eya1/Six1, Pax2, Foxg1, Fgfr1, Gata3*) are necessary for cochlear development to have not been investigated in great detail, aside from some data showing aberrant central projections in *Pax2-cre; Gata3^f/f^* mice [83].

In summary, the expression of *Lmx1a/b* is essential for proper hindbrain development, and deletion of these genes causes loss of the cochlear nucleus and projections to them. In contrast to the detailed description of *Lmx1a/b* loss, there is minimal information on the role of *Sox2.* Other central genes for cochlear nucleus development and central projections could replace lost neurons and block any auditory cochlear nucleus in *Lmx1a/b* null mice that act upstream of *Atoh1* [33,67] to repopulate lost cochlear neurons that is an age-related loss of specific nuclei.

## 5. Summary and Conclusions

The auditory system requires developing the spiral ganglion neurons that develop before hair cells and partially overlapping with auditory nuclei. SGNs depend on *Eya1* and *Sox2* to be followed by *Neurog1* and *Neurod1*, resulting in a shortening of the cochlea and smaller cochlear nuclei. Both hair cells and cochlear nuclei depend on *Eya1, Sox2,* and *Atoh1* but develop central and peripheral neuron projections to reach hair cells and cochlear nuclei in either connection. A unique dependency on *Lmx1a/b* double null mutants and the loss of SGNs, cochlear hair cells, and cochlear nuclei followed the roof plate’s loss. All transcription factors are early and require jump-start forming all three components of the auditory system that will lose hair cells and neurons. Following the *Eya1* > *Sox2* > *Atoh1* sequence of gene expression suggests the co-expression of at least three interlinked transcription factors may be needed to lead to the new formation of hair cells.

## Figures and Tables

**Figure 2 ijms-22-04189-f002:**
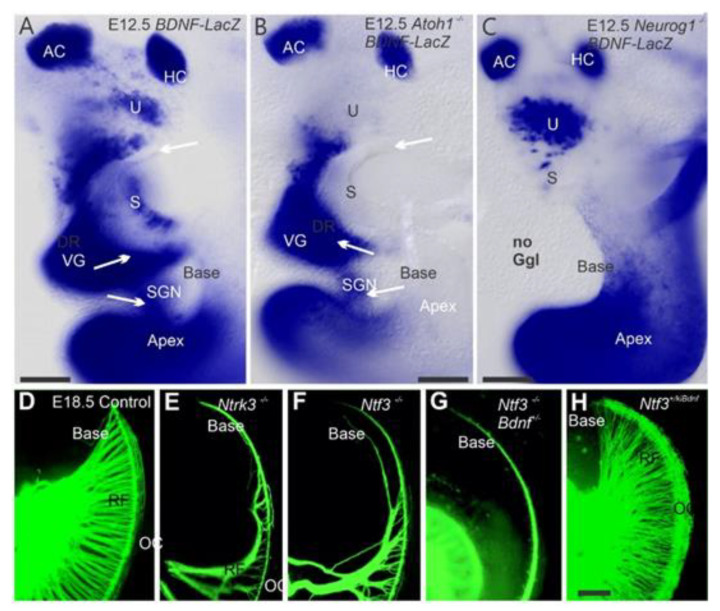
Innervation of the cochlea in various mutant mice. BDNF-LacZ of control mice (**A**) is compared with *Atoh1^f/f^; BDNF^LacZ^* (**B**) and *Neurog1^f/f^; BDNF^LacZ^* (**C**) Cochlear and vestibular ganglia (VG, SG) and hair cells are positive for *Bdnf* (**A**). Absence of *Atoh1* results in the loss of *Bdnf* expression in the utricle and saccule hair cells and the generation of fewer neurons (**B**). *Neurog1* null mutants lack all neurons but show an upregulation of *Bdnf* in the utricle hair cells (**C**). The cochlea’s innervation is fairly uniform along the entire length in P0 mice, as shown with lipophilic dye labeling (**D**). Various neurotrophin mutants show loss (**E**,**F**,**G**) and gain (**H**) of cochlear innervation. *Ntrk3* null (**E**) and *Ntf3* (NT-3) null (**F**) mutant mice show a reduction of spiral ganglion neurons innervating the basal turn of the cochlea (**E**,**F**). The reduction is more profound when *Ntf3* null is combined with haploinsufficiency for *Bdnf* (**G**)—inserting the *Bdnf* coding region into the *Ntf3* gene redirected vestibular fibers to the cochlea, increasing its overall innervation. The bar is 100 μm. Modified after [12,46,47].

**Figure 3 ijms-22-04189-f003:**
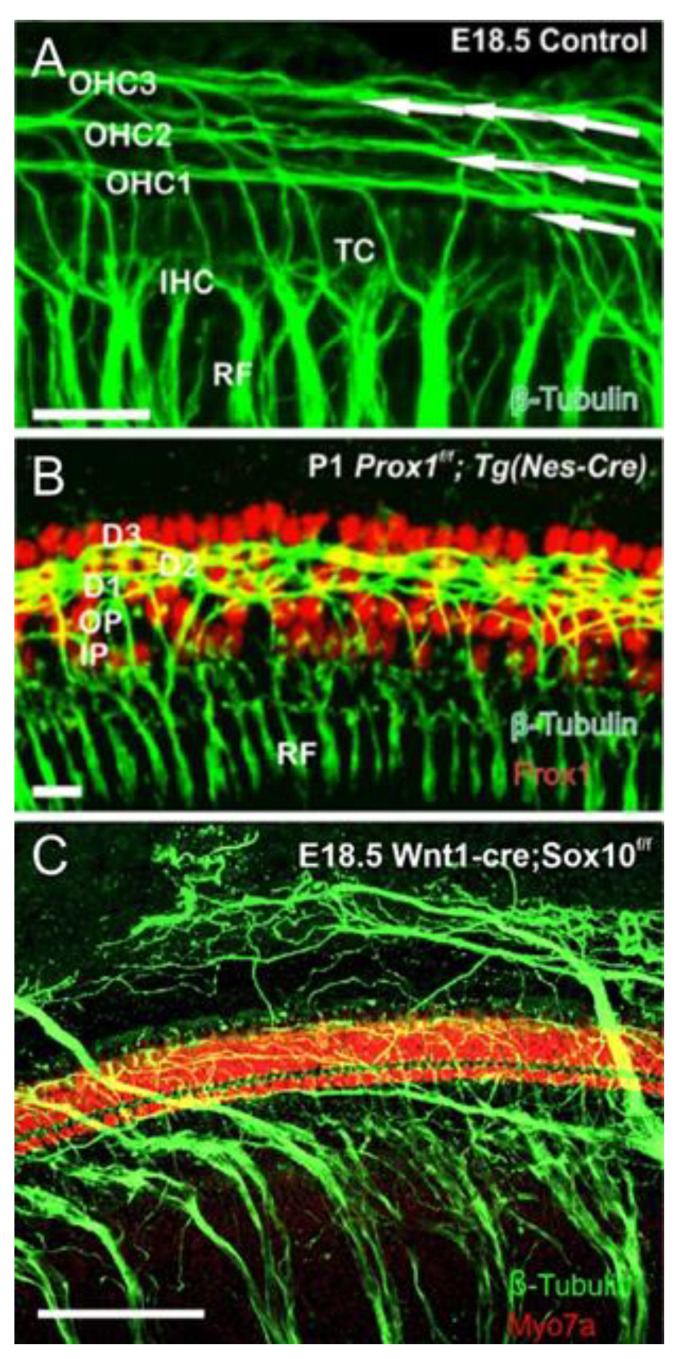
Cochlear fibers segregate by various patterns of innervation. The two types of the organ of Corti (OC) hair cells (inner hair cells, IHCs; outer hair cells, OHCs) are innervated by two types of spiral ganglion neurons (**A**). The innervation pattern shows changes in two mutations (**B**,**C**). The spiral ganglion neurons project as radial fibers (RF) to reach the hair cells in the OC. The normal innervation pattern shows a highly regular set of type I spiral ganglion neurons projecting to IHCs and type II spiral ganglion neurons, crossing the tunnel of Corti (TC) and projecting to the three rows of OHCs (**A**). Targeted deletion of *Prox1* using *Nestin-cre* (Nes-Cre) results in all tunnel crossing fibers’ aggregation between the outer pillar cells (OP; **B**). A more disruptive innervation pattern occurs with the lack of Schwann cells, as seen in *Wnt1-cre; Sox10^f/f^* mice (**C**) Radial fibers completely bypass the OC (Myo7a-positive) hair cells and expand to the lateral wall. Bars 20 μm (**A**,**B**), 100 μm (**C**). Modified after [30,46,57].

**Figure 4 ijms-22-04189-f004:**
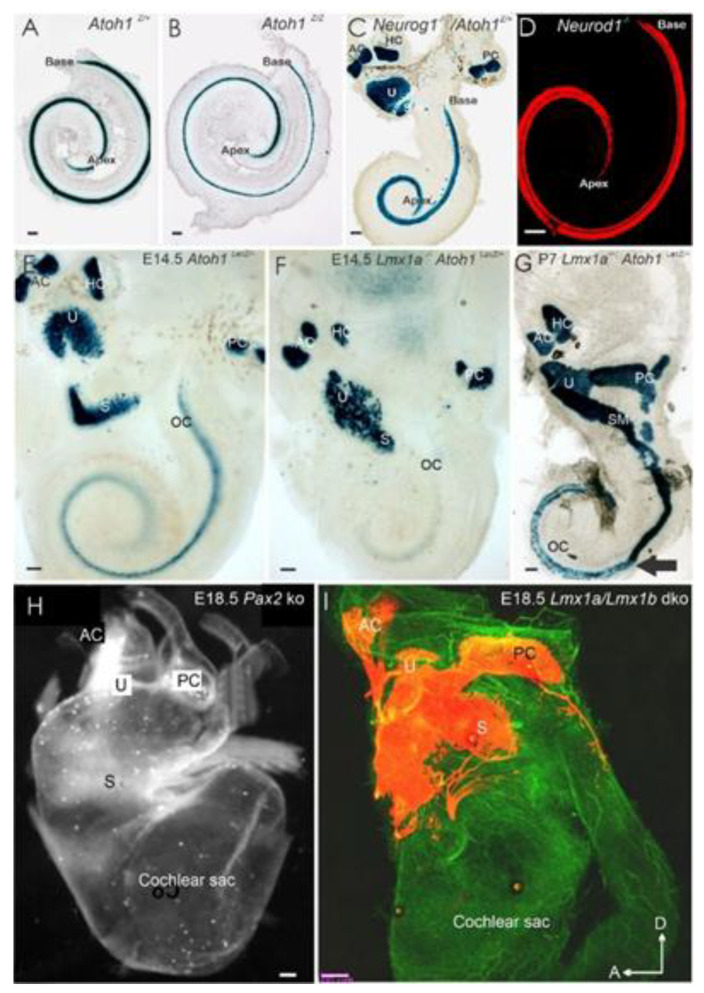
Effect of gene loss on the cochlea. Loss of *Atoh1* has a limited effect on cochlea extension (**A**,**B**) compared to *Neurog1* (**C**) and *Neurod1* (**D**), in which the cochlea is shortened. Early Atoh1^LacZ^ shows expression in hair cells near the apex (**E**) Atoh1^LacZ^ expression is delayed in *Lmx1a* null mice (**F**). Later stages of *Lmx1a* null mice show that the hair cells develop, though there is the fusion of the saccule and OC (arrow; **G**). Mice lacking *Pax2* (**H**) or *Lmx1a/Lmx1b* develop only a cochlear sac without innervation. AC, anterior crista, HC, horizontal crista; OC, organ of Corti; PC, posterior crista; S, saccule; U, utricle. A/D in I indicates dorsal and anterior directions, large arrow indicates the shift of near normal to unusual cochlear hair cells (**G**). Bar 10 µm (**A**–**H**), 100 µm (**I**) Modified after [12,66,67,68,69].

**Figure 5 ijms-22-04189-f005:**
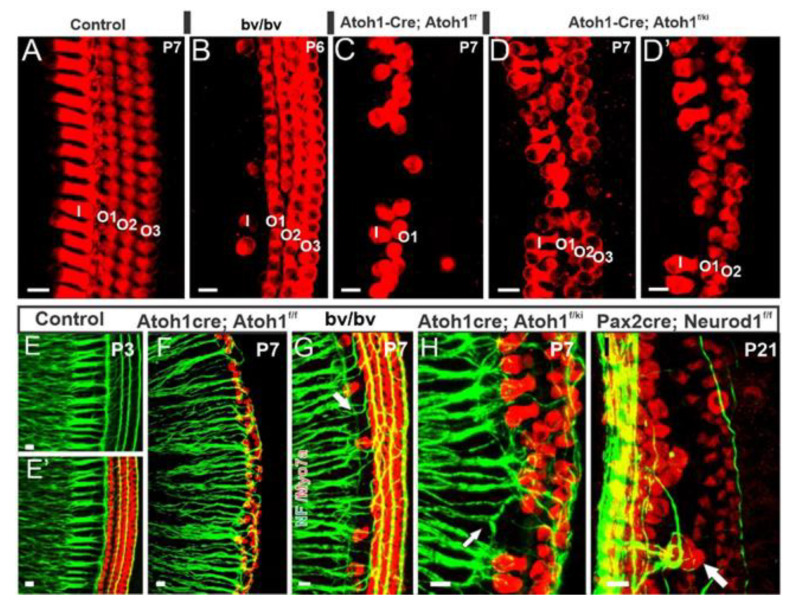
Innervation of the cochlea depends on the presence of hair cells. Detailed images are comparing hair cells and inner ear afferents in control mice (**A**), Bronx waltzer (bv/bv) (**B**), and “self-terminating” *Atoh-cre; Atoh1^f/f^* (**C**), and *Atoh-cre;Atoh1^KI/f^* (**D**,**D’**) mice. Hair cells are reduced, and inner ear innervation is disorganized in these mice. Immunolabeling for neuronal markers demonstrates the normal inner ear afferents in control mice (**E**,**E’**). This is compared with *Atoh-cre;**Atoh1^f/f^* ‘self-termination mice (**F**), Bronx waltzer (*bv/bv*) (**G**), *Atoh-cre*;*Atoh1^ki/f^* (**H**) and Pax2-cre; *Neurod1* (**I**). Bar 10 µm. Modified after [20,44].

**Figure 6 ijms-22-04189-f006:**
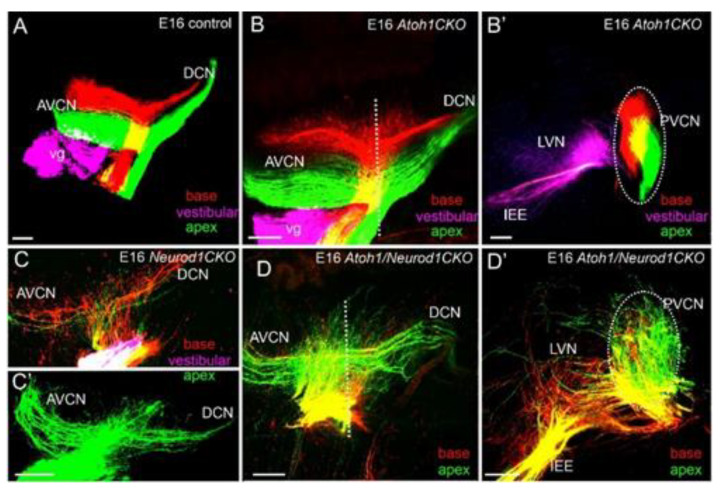
Cochlear neurons show different innervation patterns in various mutations. Central projections to the anterior (AVCN) and dorsal (DCN) cochlear nucleus are nearly identical between controls (**A**) and *Atoh1 CKO* mice (**B**,**B****’**). In contrast, both *Neurod1 CKO* (**C**,**C’**) and *Atoh1/Neurod1 CKO* mice (**D**,**D****’**) show aberrant central projections. Note the difference is obvious in control sections (**B’**) compared to *Atoh1/Neurod1 CKO* mice (**D’**). Bar 100 µm. Modified after [43].

**Figure 7 ijms-22-04189-f007:**
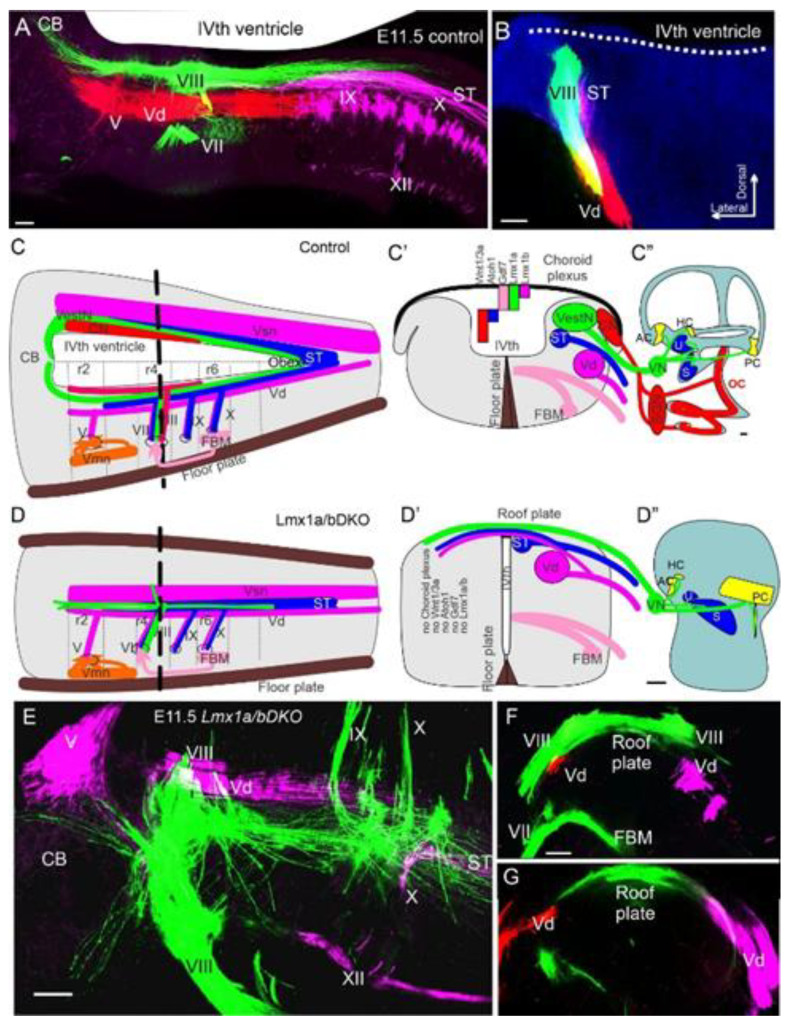
Central projections of the ear depending on the brainstem. Vestibular neurons project dorsally in the hindbrain in control and *Lmx1a/b DKO* mice (VIII; **A**–**E**). However, in *Lmx1a/b DKO* mice, central cochlear projections never develop as they do in controls (**C****”**,**D****”**). In addition, in *Lmx1a/b DKO* mice, vestibular projections interconnect across the roof plate (**D**,**D****’**,**E**,**F**,**G**), whereas vestibular fibers are normally separated by the choroid plexus (**A**,**B**,**C**,**C’**). In addition to the loss of the cochlea and spiral ganglion neurons, the cochlear nucleus does not form in *Lmx1a/b DKO* mice (**C****’**,**C****”**). Furthermore, in *Lmx1a/b DKO* mice, *Atoh1, Gdf7*, and *Wnt1/3a* expressions are absent (**C**’,**D’**). As a reference we have provide coronal section of control (**B**) and *Lmx1/a DKO* mice (**F**,**G**). Bar 100 µm. CB, cerebellum; FBM, facial branchial motoneurons; V–XII, various cranial neurons. The bar is 100 µm. Modified after [40,67].

**Figure 8 ijms-22-04189-f008:**
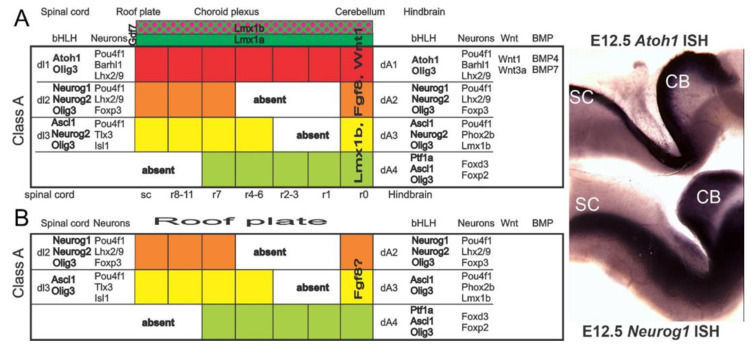
Gene expression differs in control and *Lmx1a/b* double null mice. dA1/dl1 neurons are positive for *Atoh1* and *Olig3, Pou4f1, Barhl1*, and *Lhx2/9*, extending adjacent to the roof plate from the cerebellum spinal cord shown with *Atoh1* in situ hybridization (ISH) (**A**). Note that *Lmx1a* and *Lmx1b* are upstream of *Gdf7* that disappears in *Lmx1a/b DKO* mice. *Wnt1/3a* are severely reduced, and *Atoh1* is absent in the hindbrain of *Lmx1a/b DKO* mice, suggesting the complete loss of dA1 (**B**). Note that *Neurog1* is present in r0 and contributes to part of the cerebellar neurons. Instead of a choroid plexus, we find an undifferentiated “roof plate”-like structure in a *Lmx1a/b DKO* mouse (**B**). Our *in situ* hybridization for *Atoh1* and *Neurod1* shows a segregation of these two bHLH genes (**right**). Modified from [37,40,67,126,131,137].

**Table 1 ijms-22-04189-t001:** The reduction in length of the cochlea depends on the loss of genes.

	*Atoh1* ^+/+^	*Atoh1* ^−/−^	*Neurog* ^−/−^	*Neurog1*^–/–^/*Atoh1*^−/−^	*Neurod1* ^−/−^	*Neurod1*^–/–^/*Atoh1*^−/−^
Matei et al. 2005	5.77 (100%)	5.40 (93%)	3.08 (53%)	3.00 (52%)		
Jahan et al. 2010	5.90 ± 0.4		2.45 ± 0.3		2.69 ± 0.1	
Filova et al. 2020	4.6 (E16.5)	4.5 (E16.5)			3.1 (E16.5)	2.4 (E16.5)

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
