# Peer review of "Development in the Mammalian Auditory System Depends on Transcription Factors"

_ijms, 2021, doi:10.3390/ijms22084189_

Round 1
Reviewer 1 Report
Presented manuscript should never be submitted in such early-stage form of preparation! "Fluency can be defined as writing the text in an easy-to-read manner in which no element exists causing the reader to pause while reading" (Atasoy 2016). In the case of this manuscript reading is very difficult.
Also I cannot recognize any keynote of writing -just separate parts. Conclusion and summary do not contain summary and conclusion.
Figure’ legends do not explain presented details but general description which should be presented rather in the text of manuscript.
Chapters should be numbered and instead of strange A), B) C) should be organized in 1.1, 1.2 and so on...with corresponding titles or subtitles.
But the worse is that authors even did not read what they have submitted -in manuscript still comments exist which should not be never published or presented to reviewers, for example line 73 "I recommend adding a sentence here...' or line 270 "The above green fragment is not clear"!!!
For me manuscript in current form is TOTALLY NOT CLEAR and should not be reviewed!!!
Author Response
Dear reviewer,
thank you for your detailed review, appreciated. We have now reviewed the current paper and have changed the following details:
We have reduced the description of the legends, in particular Fig. 4-8.
We have changed the numbers into 2.A, 2.B etc
Thanks
Bernd Fritzsch
Reviewer 2 Report
In this review article , Elliot et al. offer an overview of how bHLH transcription factors (TFs) regulate cell identity and neuronal patterning within the mammalian auditory system. This article is not ready for publication, first and foremost because the authors have submitted what is clearly an unfinished draft. The article is riddled with grammatical errors that severely hamper its readability. Furthermore, there are places where comments remain from another author asking for revisions (e.g. lines 73-74, 145, 271). If it were well written, this review could be a valuable resource that brings clarity to how the mammalian auditory system is regulated by a group of bHLH transcription factors. However, as it now stands, this version of the manuscript lacks such clarity. To help bring the article into focus, the following major revisions are suggested:
1. Introduction: For this article, the intro should help the reader to recognize the significance of how a small group of bHLH transcription factors interact to regulate development of the mammalian auditory system. Since the focus of this review is on bHLH genes in the auditory system, the reader would be well served with an intro that includes some relevant basic info about bHLH TFs and how they function. Additional helpful information includes a description of the auditory system and what each part does, along with an overview of the temporal progression of it development (which the authors do provide).
2. Subject matter: Ideally, the article would review experimental findings and synthesize them in such a way as to provide new insights into an existing body of knowledge. The title promises a review on bHLH transcription factors in the auditory system, but omits to mention that a fair bit of space will be dedicated to non-bHLH TFs, and indeed, non-TFs, and their role in patterning the auditory system. Occasionally, these other factors are discussed in relation to NeuroG1, NeuroD1, and Atoh1, but more often than not, the non-bHLH factors are reviewed without any explicit connection drawn with the core bHLH factors. This makes the manuscript read more like an unwieldy overview of auditory system development rather than a focused work. In the interest of keeping this article focused, manageable, and informative, I recommend removing or revising the sections where the non-bHLH factors (i.e. sox2, lmx1a/b, gata3, dicer, ntf3, bdnf, wnt3, pax2, eya1, etc.) are not explicitly linked in with the central theme of the article. Of course, where there are demonstrable connections between a bHLH transcription factor and non-bHLH factors, these should be emphasized. Also, when a new gene or factor is first mentioned, it would be helpful to specify what it is, i.e. TF, ligand, receptor, etc.
3. Organization of the manuscript: The article would benefit from a structural reorganization to minimize unnecessary repetition and to help clarify the narrative. As it stands now, the main body of the manuscript begins by reviewing some temporal aspects of how some bHLH transcription factors instruct the development of the 3 main parts of the mammalian auditory system: A) spiral ganglion, B) cochlear hair cells, and C) cochlear nuclei. This section is followed by one where the authors go on to repeat and expand upon some of the info in A, B, and C, but his time the headings are alphanumerical (e.g. 1.A), 1.B), 2.A), etc): the numeric portion corresponds to the previously described anatomical structures in the auditory system and the alpha portion refers to a specific or group of transcription factors (A - Atoh1, B - NeuroG1 and NeuroD1 regulation of Atoh1, C - Sox2 and other factors). The benefit to this organizational strategy is unclear since it creates a convoluted narrative and encourages unnecessary repetition of info. To streamline the manuscript, the authors may consider integrating section A with sections 1.A-C, section B with sections 2.A-C and section 3 with sections 3.A-C, using appropriate subheadings where necessary.
4. Summary and conclusion: More emphasis on unanswered questions uncovered in the review and some indication of where the field is headed is recommended.
5. Detailed revisions:
- I leave it to the authors to identify and correct the grammar, spelling, and punctuation mistakes that are throughout the manuscript.
- Line 30: should read "The mammalian auditory system" since most auditory systems do not share these derived traits
- Line 60: legend should indicate that the images are of GFP driven by alpha9-AChr regulatory elements
- Lines 72 and 81: how can alpha9-AChr and p27kip both be markers of HC differentiation? Given p27kip's role in regulating the cell cycle, does it make more sense to call p27 a marker of cell cycle exit? It may also make sense to reorganize this paragraph to discuss cell cycle exit first, followed by HC differentiation.
- Line 83: beta-gal stain is not typically an immunochemical reaction
- line 143: Brn3c and Pou4f3 are two names for the same transcription factor.
- line 153: the meaning of this sentence is completely opaque.
- lines 179-197: if the authors wish to include these "other" factors, they should be under the appropriate heading (1.C)
- lines 282-285: Meaning is unclear. What "various ways"? Which "Atoh1 deletions"? Do Atoh1 deletions lead to more apical HCs, even though Atoh1 is absolutely required for HC development? "Increased expression" of what? How can undifferentiated HCs be identified?
- Line 287: Myo7aa is very commonly used as a marker for differentiated HCs. How can myo7aa-positive HCs be undifferentiated?
- Lines 300-302: How are these findings related to Atoh1 function?
- lines 305-309: These sentences on regeneration seem out of context here.
Author Response
Dear reviewer,
thank you so much for suggesting the current submission. Please find attached a seriously changed original submission that is now improved.
1 + 2. We reorganize the Introduction, shorten the description.
3. We take out the A,B,C in the introduction and followed your suggested to use 2.A., 2.B and 2.C to streamline our presentation. Thank you for the suggestion.
4. We eliminated a final summary but expanded the individual headings.
5. Review is now clear and has seen by all authors.
- We front load with the mammalian auditor system.
- Reorganized the alpha 9 at a later stage after p27.
- We switched to Pou4f3 throughout.
- A phrase was corrected.
- Indeed, we have included additional citations are include in C for all.
- Rephrase an unclear in the original submission, now clarified.
- Myo7a is only used as a marker.
- Atoh1 is now clearer in the final version
Thank you our detailed review which we are looking forward to submit the final paper.
Round 2
Reviewer 1 Report
The 'revised' version should be prepared as tracking changes and IJMS template should be used. In current form pages numbering cannot be used, all pages are just 1! Manuscript stil is not organized properly.
Additionally authors were not replaying to my comments. Just were writing 2 short sentences without detailed description of changes.
This manuscript is not appropiate in current form for publishing and authors do not follow reviewer suggestions to improve it.
Author Response
The 'revised' version should be prepared as tracking changes and IJMS template should be used. In current form pages numbering cannot be used, all pages are just 1! Manuscript stil is not organized properly.
Additionally authors were not replaying to my comments. Just were writing 2 short sentences without detailed description of changes.
This manuscript is not appropiate in current form for publishing and authors do not follow reviewer suggestions to improve it.
Dear reviewer,
I am sorry, we were under pressure to resubmit the paper. We have now followed the outline, have provided numbers, highlighted the changes in red bars and have updated our paper.
In details, we have changed the title, have expanded the abstract and introduction to set the stage for the interaction of transcription factors in the auditory system. We also added a Summary and Conclusion that is a condense of our insights. Finally, we added the gene acronyms to help follow the narrative.
Thank you for your help, we hope the final review will be improved by your suggestions.
Reviewer 2 Report
The authors have improved this manuscript relative to the unfinished draft that was originally submitted for review. Unfortunately, many of the required revisions from my previous evaluation remain unfixed in this version and further revisions are required to get the manuscript into a publishable form.
1. The title on the MDPI site "Cochlear innervation, hair cells, and cochlear nuclei: the role of bHLH genes in the connections" does not match the current title of the manuscript "Cochlear innervation, hair cells, and cochlear nuclei define the mammalian auditory system". I would argue than neither title is suitable for this review. The original title puts too much emphasis on bHLH factors when in fact the review covers a great number of TFs (bHLH and non-bHLH) and neuropeptides that pattern the mammalian auditory system. The current title doesn't adequately describe the content of the review and doesn't really make sense (the individual parts of the auditory system define the whole system?)
2. "Introduction and historical perspective" - It's not clear what the purpose of this section is. There's not much "historical perspective" and most (if not all) of the info gets repeated in the subsequent sections. Additionally, if a relative newcomer to the auditory system came upon this review, that poor reader would have no idea how HCs, SGNs, and the CN function to produce the thing we know as auditory sensory perception. I recommend giving the Intro a reason to exist by providing a brief overview of the auditory system and how it works.
3. Line 78: "A different gene, p27(kip)..." - different from what?
4. Line 80: "a marker of differentiation" - No, p27(kip) is not a marker of HC differentiation. p27 is a marker of cell cycle exit.
5. Lines 96-97: Unclear wording (what's an "early gene expressions"?). Also, it's worth pointing out that base-apex expression of alpha9 is consistent with the base-apex expression of Atoh1.
6. Line 103: should read something like "...was overstated as it was *subsequently* shown that..."
7. Lines 143-145: The lack of HCs (intraganglionic or not) in atoh1/neurod1 double mutants merely indicates that Atoh1 acts downstream of NeuroD1 in HC specification.
8. Lines 156-157: SGNs can be classified as *either* Type I or Type II neurons, not both.
9. Lines 196-198: meaning unclear
10. Lines 213-214: awkward wording
11. Line 217: what's the condition of the conditional deletion?
12. Lines 224-225: meaning unclear
13. Line 230: should read "Sox2 *is* a critical gene..."
14. Lines 240-241: Either delete the word "negatively" or qualify the sentence to say that *mutations* in genes have negative effects on the development of afferents.
15. Line 247: Similar comment to above for lines 240-241. I think the author is trying so say that mutations in genes derail the process?
16. Line 248: there are a lot of neurotrophins. Which ones are relevant here?
17. Line 264: "upstream" of what?
18. Lines 280-281: I agree with whoever made the original comment - the previous sentence is not clear. Also, the authors should really do better to submit their best, curated work.
19. Lines 293-296: This section doesn't make sense. What is a "cellular progression"? Which Atoh1 deletions? How are they different from a simple Atoh1 deletion? How are HCs formed in Atoh1 mutants? How can you identify an undifferentiated cell as an IHC or an OHC (which are both differentiated cell types)?
20. Line 299: Myo7a is a marker of differentiated HCs.
21. Line 303: should read "*These* data..."
22. Line 312: can the authors clarify what is meant by "upper limit of proliferation"?
23. Lines 312-319: How is this info about innervation and non-Atoh1 genes related to the subject of this paragraph (Atoh1)?
24. Line 312: "certain mutations" - since this entire paragraph was about Atoh1, it would be helpful if the authors qualified this statement to clarify that they are referring to genes other than Atoh1. Also, the authors go on to describe only a single gene (Srrm4) which appears to affect a subset of HCs when mutated. So rather than say "certain mutations", just say Srrm4.
25. Line 313: Since this is the first time Srrm4 is mentioned, it would be helpful to explain what Srrm4 is.
26. Line 313-314: awkward wording "loss of IHCs...lack most IHCs"
27. Lines 315-316: Is this "important"? The reader hasn't been told anything about either Srrm4 or REST, so why should the reader be impressed by this finding? Also, lines 313-314 just said that Srrm4 mutants lack most IHCs. Does ectopic expression of DN-REST exacerbate this phenotype in Srrm4 mutants?
28. Line 316: typo "IHC/s"
29. Line 374: Sox2 is upstream of which bHLH genes?
30. Line 412: typo "cochleare"
31. Line 433: should read "cochlear"
32. Line 450-452: If Atoh1 has no apparent role in "central projections' guidance" (awkward phrasing), why is more work needed?
33. Line 455: awkward - "expression of NeuroG1 expression"
34. Line 463-464: DCN requires NeuroD1 for what?
35. Line 495: "other genes' expression" should read "the expression of other genes"
36. Line 501: should read "...projections *in mutants for* other genes..."
37. Line 503: delete the comma between "data" and "showing"
38. From the authors rebuttal: "We eliminated a final summary but expanded the individual headings." I disagree with the authors' choice to omit a final summary.
39. General comment - given the large number of acronyms and gene names used throughout the manuscript, it is required that the authors include a "List of Gene Acronyms and Abbreviations" section in the manuscript.
Author Response
The authors have improved this manuscript relative to the unfinished draft that was originally submitted for review. Unfortunately, many of the required revisions from my previous evaluation remain unfixed in this version and further revisions are required to get the manuscript into a publishable form.
Thank you for the positive review, appreciated. We have no time out rethink the paper and has a revision ready for you. Importantly, we have changed the title, have added on overview of the introduction and have added a summary and conclusion. Thank you for suggesting the acronyms, they are now attached.
- The title on the MDPI site "Cochlear innervation, hair cells, and cochlear nuclei: the role of bHLH genes in the connections" does not match the current title of the manuscript "Cochlear innervation, hair cells, and cochlear nuclei define the mammalian auditory system". I would argue than neither title is suitable for this review. The original title puts too much emphasis on bHLH factors when in fact the review covers a great number of TFs (bHLH and non-bHLH) and neuropeptides that pattern the mammalian auditory system. The current title doesn't adequately describe the content of the review and doesn't really make sense (the individual parts of the auditory system define the whole system?)
Indeed, we have final ideas for the title but have now our new title. We also had a recent, important paper (Xu et al., PNAS, 2021) that shows that Eya1 is upstream from Sox2 and all ear related bHLH genes will depend on them. We leave the same organization but have expandet the role of Eya1. Thank you.
- "Introduction and historical perspective" - It's not clear what the purpose of this section is. There's not much "historical perspective" and most (if not all) of the info gets repeated in the subsequent sections. Additionally, if a relative newcomer to the auditory system came upon this review, that poor reader would have no idea how HCs, SGNs, and the CN function to produce the thing we know as auditory sensory perception. I recommend giving the Intro a reason to exist by providing a brief overview of the auditory system and how it works.
We have added a novel introduction to expand the broad perspective of our approach.
- Line 78: "A different gene, p27(kip)..." - different from what?
We have rephrased the sentence. (Please highlight are marked in red, like all the other comments)
- Line 80: "a marker of differentiation" - No, p27(kip) is not a marker of HC differentiation. p27 is a marker of cell cycle exit.
Sorry, we related the marker but have now clarified our use in cell cycle exit, thank you.
- Lines 96-97: Unclear wording (what's an "early gene expressions"?). Also, it's worth pointing out that base-apex expression of alpha9 is consistent with the base-apex expression of Atoh1.
Yes, pointed this fact out at several times, thank your for pointing this out. We have rephrased the line
- Line 103: should read something like "...was overstated as it was *subsequently* shown that..."
We have added ‘subsequently’, thank you.
- Lines 143-145: The lack of HCs (intraganglionic or not) in atoh1/neurod1 double mutants merely indicates that Atoh1 acts downstream of NeuroD1 in HC specification.
Indeed, we showed the expression of transcription factors that have a unique expression in the ganglion, including Atoh1, Fgf8 and Nhlh1.
- Lines 156-157: SGNs can be classified as *either* Type I or Type II neurons, not both.
Thank you, we clarified that sentence.
- Lines 196-198: meaning unclear
We have rephrased the sentence.
- Lines 213-214: awkward wording
Thank you, we have changed the wordings.
- Line 217: what's the condition of the conditional deletion?
Sorry, a lapsus lingua. We have changed the wordings.
- Lines 224-225: meaning unclear
We have reworded the entire sections, hope the clarification helped?
- Line 230: should read "Sox2 *is* a critical gene..."
Indeed, since we added Eya1 and Sox2 as they are upstream of bHLH genes.
- Lines 240-241: Either delete the word "negatively" or qualify the sentence to say that *mutations* in genes have negative effects on the development of afferents.
We changed the wording, adding of ‘mutations’, as suggested.
- Line 247: Similar comment to above for lines 240-241. I think the author is trying so say that mutations in genes derail the process?
Yes, genes derail the process, thank for the suggestion.
- Line 248: there are a lot of neurotrophins. Which ones are relevant here?
Two genes are important, Bdnf and Ntf3. Double null of both results in the absence of all neurons (Kersigo and Fritzsch, 2015).
- Line 264: "upstream" of what?
We changed the wording.
- Lines 280-281: I agree with whoever made the original comment - the previous sentence is not clear. Also, the authors should really do better to submit their best, curated work.
Sorry, this oversight was depending on using different colors used by each of the authors. Unfortunately, this author did not switch on the color. We have now rephrased text 280-281.
- Lines 293-296: This section doesn't make sense. What is a "cellular progression"? Which Atoh1 deletions? How are they different from a simple Atoh1 deletion? How are HCs formed in Atoh1 mutants? How can you identify an undifferentiated cell as an IHC or an OHC (which are both differentiated cell types)?
All HC depend on Atoh1. However, at which time Atoh1 is deleted and how much undifferentiated HCs make part of our work. We hope the rephrasing is fine with the sentences of 293-296.
- Line 299: Myo7a is a marker of differentiated HCs.
Yes, it is upregulated late in embryos after E14.5.
- Line 303: should read "*These* data..."
Thank you, corrected.
- Line 312: can the authors clarify what is meant by "upper limit of proliferation"?
We have rephrased the sentence, hope the clarity is improved now?
- Lines 312-319: How is this info about innervation and non-Atoh1 genes related to the subject of this paragraph (Atoh1)?
We have rewritten this sentence and have a separation to detail with Srrm3 (line 312). Thank you for pointing up the problem, we hope to clarify line 313-319.
- Line 312: "certain mutations" - since this entire paragraph was about Atoh1, it would be helpful if the authors qualified this statement to clarify that they are referring to genes other than Atoh1. Also, the authors go on to describe only a single gene (Srrm4) which appears to affect a subset of HCs when mutated. So rather than say "certain mutations", just say Srrm4.
Thank you, we have rephrased the previous part of line 312.
- Line 313: Since this is the first time Srrm4 is mentioned, it would be helpful to explain what Srrm4 is.
We expanded how Srrm3/4 interact with REST to ablate all IHCs.
- Line 313-314: awkward wording "loss of IHCs...lack most IHCs"
Rephrased, thank you.
- Lines 315-316: Is this "important"? The reader hasn't been told anything about either Srrm4 or REST, so why should the reader be impressed by this finding? Also, lines 313-314 just said that Srrm4 mutants lack most IHCs. Does ectopic expression of DN-REST exacerbate this phenotype in Srrm4 mutants?
REST interacts with Srrm3/4 and blocks early IHC after around E18.
- Line 316: typo "IHC/s"
Corrected.
- Line 374: Sox2 is upstream of which bHLH genes?
Rephrased by adding Eya1 and Sox2 are needed for bHLH genes in the ear.
- Line 412: typo "cochleare"
Corrected.
- Line 433: should read "cochlear"
Corrected
- Line 450-452: If Atoh1 has no apparent role in "central projections' guidance" (awkward phrasing), why is more work needed?
Loss of Atoh1 deletes cochlear nuclei. Interestingly enough, the central projection happens in the absence of cochlear nuclei formation. We rephrased sentence 450-452.
- Line 455: awkward - "expression of NeuroG1 expression"
Rephrased.
- Line 463-464: DCN requires NeuroD1 for what?
DCN is highly positive for Neurod1 that is early on downregulated in Atoh1. We have an upregulation of Atoh1 in the absence of Neurod1, suggesting a counteract of both bHLH genes.
- Line 495: "other genes' expression" should read "the expression of other genes"
Thank you, corrected.
- Line 501: should read "...projections *in mutants for* other genes..."
Thank you, we corrected the sentence.
- Line 503: delete the comma between "data" and "showing"
Thanks, corrected.
- From the authors rebuttal: "We eliminated a final summary but expanded the individual headings." I disagree with the authors' choice to omit a final summary.
We have provided a summary and conclusion that allows us to provide a broader perspective.
- General comment - given the large number of acronyms and gene names used throughout the manuscript, it is required that the authors include a "List of Gene Acronyms and Abbreviations" section in the manuscript.
Thank you for the suggestions, we added a list of acronyms.
Round 3
Reviewer 1 Report
Manuscript in current form has been significantly improved in comparison to previously submitted versions. However, some inadequacy still exist.
1) Title is too general. Obviously transription factors have role in development...It should be precised.
2) Figures still have no evident titles. Panel legends should contain first (A) and then description and so on....
It is very difficult to follow panels explanations. It must be improved.
3) Importantly, Figure 2, 3, 4, 5 legends do not correspond to Figures 2, 3, 4, 5 images!
4)bHLH proteins are transcription factors as well (p. 1 line 13/14)
5) I believe that 'genes coding bHLH proteins' would sound better than 'bHLH genes' but this is suggestion....
6) How genes interact directly? Rather proteins interact or expressed from activated gene protein can impact other gene...
7) p8 line 251 in the sentence 'Additional genes, besides Sox2, have been shown negative effects on the develop-251 ment of inner ear afferents" something is missing?
8) p8line273 again genes interact?-
Author Response
Thank you for reviewing the paper, appreciated. We have no fine tuned the following changes, highlighted in red.
1) Title is too general. Obviously transription factors have role in development...It should be precised.
New title: Transcription factors depend on development in the mammalian auditory system.
2) Figures still have no evident titles. Panel legends should contain first (A) and then description and so on....
It is very difficult to follow panels explanations. It must be improved.
We added short introductions to summarize the figures
3) Importantly, Figure 2, 3, 4, 5 legends do not correspond to Figures 2, 3, 4, 5 images!
Sorry, we try to follow the sequence, now they follow strictly the sequence.
4)bHLH proteins are transcription factors as well (p. 1 line 13/14)
New first line is: We review the molecular basis of several transcription factors (Eya1, Sox2) including the three related genes coding basic Helix-loop-Helix (bHLH) proteins (Neurog1, Neurod1, Atoh1) to develop spiral ganglia, cochlear nuclei, and cochlear hair cells.
5) I believe that 'genes coding bHLH proteins' would sound better than 'bHLH genes' but this is suggestion...
Thank you, we changed the wording in most cases, starting with the abstract.
6) How genes interact directly? Rather proteins interact or expressed from activated gene protein can impact other gene...
Yes, they do in part interact and in part as directly.
7) p8 line 251 in the sentence 'Additional genes, besides Sox2, have been shown negative effects on the develop-251 ment of inner ear afferents" something is missing?
We eliminated this sentence, starting at: For instance, Eya1…..
8) p8line273 again genes interact?-
Reworded: …gene proteins interact with each other…